# Iron Deficiency and Reduced Muscle Strength in Patients with Acute and Chronic Ischemic Stroke

**DOI:** 10.3390/jcm11030595

**Published:** 2022-01-25

**Authors:** Nadja Scherbakov, Anja Sandek, Miroslava Valentova, Antje Mayer, Stephan von Haehling, Ewa Jankowska, Stefan D. Anker, Wolfram Doehner

**Affiliations:** 1Berlin Institute of Health, Center for Regenerative Therapies (BCRT), Charité—Universitätsmedizin Berlin, 10117 Berlin, Germany; antje.mayer@charite.de (A.M.); stefan.anker@charite.de (S.D.A.); wolfram.doehner@charite.de (W.D.); 2Center for Stroke Research Berlin (CSB), Charité—Universitätsmedizin Berlin, 10117 Berlin, Germany; 3Department of Internal Medicine and Cardiology, Campus Virchow-Klinikum, Charité—Universitätsmedizin Berlin, 10117 Berlin, Germany; 4German Centre for Cardiovascular Research (DZHK), Partner Site Berlin, 10785 Berlin, Germany; 5Department of Cardiology and Pneumology, University of Göttingen, 37073 Göttingen, Germany; anja.sandek@med.uni-goettingen.de (A.S.); Miroslava.Valentova@med.uni-goettingen.de (M.V.); stephan.von.haehling@med.uni-goettingen.de (S.v.H.); 6German Centre for Cardiovascular Research (DZHK), Partner Site Göttingen, 37075 Göttingen, Germany; 7Institute of Heart Disease, Wroclaw Medical University, 50-367 Wroclaw, Poland; ewa.jankowska@umed.wroc.pl; 8Institute of Heart Disease, University Hospital, 50-367 Wroclaw, Poland; 9Division of Cardiology and Metabolism-Heart Failure, Cachexia & Sarcopenia, Department of Cardiology (CVK), Charité-Universitätsmedizin Berlin, 10117 Berlin, Germany

**Keywords:** iron deficiency, prevalence, acute ischemic stroke, chronic stroke, muscle strength, functional outcome

## Abstract

(1) Introduction: Iron deficiency (ID) contributes to impaired functional performance and reduced quality of life in patients with chronic illnesses. The role of ID in stroke is unclear. The aim of this prospective study was to evaluate the prevalence of ID and to evaluate its association with long-term functional outcome in patients with ischemic stroke. (2) Patients and Methods: 140 patients (age 69 ± 13 years, BMI 27.7 ± 4.6 kg/m², mean ± SD) admitted to a university hospital stroke Unit, with acute ischemic stroke of the middle cerebral artery were consecutively recruited to this observational study. Study examinations were completed after admission (3 ± 2 days after acute stroke) and at one-year follow up (*N* = 64, 382 ± 27 days after stroke). Neurological status was evaluated according to the National Institute of Health Stroke Scale (NIHSS) and the modified Rankin scale (mRS). Muscle isometric strength of the non-affected limb was assessed by the maximum handgrip test and knee extension leg test. ID was diagnosed with serum ferritin levels ≤ 100 µg/L (ID Type I) or 100–300 µg/L if transferrin saturation (TSAT) < 20% (ID Type II). (3) Results: The prevalence of ID in acute stroke patients was 48% (*N* = 67), with about two-thirds of patients (*N* = 45) displaying ID Type I and one-third (*N* = 22) Type II. Handgrip strength (HGS) and quadriceps muscle strength were reduced in patients with ID compared to patients without ID at baseline (HGS: 26.5 ± 10.4 vs. 33.8 ± 13.2 kg, *p* < 0.001 and quadriceps: 332 ± 130 vs. 391 ± 143 *N*, *p* = 0.06). One year after stroke, prevalence of ID increased to 77% (*p =* 0.001). While an improvement of HGS was observed in patients with normal iron status, patients with ID had no improvement in HGS difference (4.6 ± 8.3 vs. −0.7 ± 6.5 kg, *p* < 0.05). Patients with ID remained with lower HGS compared to patients with normal iron status (28.2 ± 12.5 vs. 44.0 ± 8.6 kg, *p* < 0.0001). (4) Conclusions: Prevalence of ID was high in patients after acute stroke and further increased one year after stroke. ID was associated with lower muscle strength in acute stroke patients. In patients with ID, skeletal muscle strength did not improve one year after stroke.

## 1. Introduction

Stroke is one of the leading causes of disability in adult life with a global annual incidence rate over 12 million cases [1]. Clinical outcome after stroke depends among other factors on the presence of comorbidities, such as hypertension, diabetes mellitus, heart failure (HF), or chronic kidney disease (CKD) [2,3,4]. Growing evidence suggests a considerable impact of iron deficiency (ID) on clinical course, prognosis, and quality of life in geriatric patients, as well as in patients with chronic diseases, such as chronic HF, cancer, and CKD [5,6,7,8].

A well-balanced iron metabolism plays a central role in the maintenance of numerous biological processes, including erythropoiesis, oxidative metabolism, immune response, and neurotransmission [9]. Physiologically, the body absorbs 7–10% of dietary iron per day, which suggests that malnutrition and/or malabsorption may have a major impact on iron balance [10]. In chronic inflammatory conditions, sufficient absorption of iron by the gastrointestinal tract and cellular iron export from the body iron stores is inhibited, leading to development of functional ID [11,12]. Inadequate nutritional iron uptake or absorption, as well as excessive blood loss, are the main causes leading to the development of absolute ID [13]. Biochemically, ID is manifested when the extracellular iron of the bone marrow, ferritin plasma levels, and transferrin saturation are low [13].

Up to date, a presence of ID, ID type, and its impact on clinical outcome remained not sufficiently studied in stroke. We hypothesized that ID is associated with low functional performance in patients with stroke. In this observational study, we aimed to evaluate the prevalence of ID in patients with acute stroke and at one year after stroke. Additionally, functional outcome in relation to ID after acute stroke and at one year after the event was investigated.

## 2. Methods

### 2.1. Study Design and Population

We prospectively studied 140 patients with acute ischemic stroke (AIS) in the territory of the middle cerebral artery (MCA), participating in the longitudinal prospective observational Body Size in Stroke Study [14] (BoSSS, German registry for clinical trials number DRKS00000514). The patients with mild to moderate neurological deficit (defined by the National Institute of Health Stroke Scale (NIHSS) as ≤12 points) were consecutively enrolled within 48 h after stroke onset, being admitted to the Stroke Unit at a tertiary university center (Charité University Hospital Berlin, Campus Virchow Clinic, Berlin, Germany).

Stroke was classified according to Trial of ORG 10172 in Acute Stroke Treatment (TOAST) classification, etiology of cardio embolism, large-artery atherosclerosis, small-vessel occlusion, and stroke of undetermined etiology [15]. One year after stroke, patients were invited to a follow up examination (Follow up cohort, FU cohort).

The study was approved by the Ethics Committee of Charité University Hospital Berlin (EA2/008/09), and all patients gave written informed consent.

### 2.2. Assessment of Functional Status

Study examinations were performed at baseline (3 ± 2 days after acute stroke) and at one-year follow up (FU, mean 382 ± 27 days after stroke). Functional status was assessed by the modified Rankin Scale (mRS), which measures physical independency by assessment of the body function, activity, and participation in daily tasks on the scale ranging from “0” (no symptoms) to “6” (death) [16].

Short physical performance battery (SPPB) was performed to assess functional capacity. SPPB includes examinations of standing ability with both feet together in the side-by-side, semi-tandem, and tandem positions; 4-m walk; and time taken to rise five times from the chair and return to the seated position. The score ranges from 0 (not attended) to 12 points (completed) [17].

Isometric muscle strength of the hand was assessed by the handgrip strength (HGS) test using a handgrip dynamometer (Saehan Corporation, Changwon, Korea). The highest of three handgrip measurements of the non-paretic or strongest arm was used for analysis. HGS below mean was considered as low muscle strength (low HGS).

Maximal isometric muscle strength of the quadriceps muscle (Newton, *N*) was measured as described previously [18]. Briefly, the freely hanging legs of the sitting patients were connected at the ankle with a pressure transducer (Multitrace 2, Lectromed, Jersey, Channel Islands), and maximal isometric strength was assessed from the best of three contractions on each leg, with a resting period of at least 60 s in between.

### 2.3. Body Composition and Nutritional Status

Appetite (the subjective desire to eat) was assessed according to the visual analogue scale (VAS) ranging from “0” (no appetite at all) to ”10” (always a very good appetite) [19]. None of the patients included in our study had dysphagia on a clinically relevant level (preventing oral feeding) and none were fed enterally or parenterally. The nutritional status was assessed by Mini Nutritional Assessment (MNA) at 12 months as follows: normal nutritional status was considered if the patients achieved ≥24 points, and all other patients were considered to have low nutritional status [20].

The body mass index (BMI) was calculated as a ratio of body weight to height squared (kg/m^2^). 

### 2.4. Blood Sampling and Iron Deficiency

Venous blood samples we obtained in all patients after 12 h of overnight fasting. Standard biochemical parameters were assessed by routine laboratory measurement. Normal iron status was defined by a serum ferritin > 100 µg/mL and transferrin saturation (TSAT) ≥ 20% [21]. Iron deficiency type I (ID I) was considered when plasma ferritin levels were ≤100 µg/mL, and iron deficiency type II (ID II) was considered for plasma ferritin levels of 100–300 µg/L and TSAT < 20% [22]. Anemia was defined by hemoglobin plasma levels of fewer than 12 g/L in females and 13 g/L in males [13]. Systemic inflammation was present if the C-reactive protein (CrP) plasma level was over 6.1 mg/dL, as defined previously [23].

### 2.5. Statistical Analysis

All data are presented as means ± standard deviation (SD), median [interquartile range (IQR)], or percentage as appropriate. All variables were tested for normal distribution using the Kolmogorov-Smirnov test. Serum levels of CrP were non-normally distributed, and statistical comparisons between subgroups were made using analysis of variance (ANOVA), followed by Fisher’s post hoc test, Mann–Whitney or Kruskal–Wallis test, or analysis of covariance (ANCOVA). The Chi square test was used to assess categorical distribution between the groups. Multivariable models for associations of risk factors with low HGS were applied (logistic regression analysis), including all factors showing a *p*-value ≤ 0.1 in univariable analysis. Furthermore, age and BMI were added. Odds ratios with 95% confidence intervals (OR [95% CI]) were reported. A value of *p* < 0.05 was considered statistically significant. A total of 76 patients (54%) without data at follow up were excluded from follow-up analyses. Statistical analyses were performed with the StatView 5.0 software package (SAS Institute Inc., Cary, NC, USA).

## 3. Results

### 3.1. Baseline

Baseline clinical characteristics of all patients with acute ischemic stroke (*N* = 140) are presented in Table 1. Study examinations were performed at mean 3 ± 2 days after stroke.

#### 3.1.1. Iron Status at Baseline

Iron deficiency was present in 67 patients (48%) (Table 1, Figure 1A). ID was more frequently observed in women (57%) compared to men (43%, *p* < 0.001). There were no significant differences regarding clinical characteristics, medication, and frequency of comorbidities, except anemia (Table 1). Patients with and without ID showed a similar severity of neurologic deficit after stroke, as indicated by the NIHSS scale, and functional dependency, as indicated by the mRS.

ID type I was found in 32% of patients, and ID type II was found in 16% of patients. Patients with ID II were older, more frequently had a cardioembolic type of stroke, and more frequently had systemic inflammation and elevated white blood cell counts compared to patients with ID type I and patients with normal iron status (Table 1).

#### 3.1.2. Physical Status at Baseline

At baseline, patients with ID showed lower maximal HGS compared to patients with normal iron status (mean 26.5 ± 10.4 vs. 33.8 ± 13.3 kg, *p* < 0.001). In subgroup analysis, the lower HGS was observed in patients with ID type II, followed by patients with ID type I vs. patients with Normal Iron (mean 23.0 ± 8.8 vs. 28.0 ± 10.7 vs. 33.8 ± 13.3 kg, *p* < 0.01, Figure 2).

Of all patients, 78 patients (56%) were able to perform the quadriceps strength test. Patients with ID showed a trend towards lower quadriceps muscle strength compared to the patients with normal iron status (mean 332 ± 130 vs. 391 ± 143 *N*, *p* = 0.06) independently of their type of ID (I or II) (not shown).

SPPB was performed in 50% of patients with ID and with normal iron status. There was no significant difference either in frequency of attendance to the SPPB or in score values between both subgroups (Appendix A Appendix A).

#### 3.1.3. Nutritional Status at Baseline

At baseline, there was no difference in self-reported appetite according to the visual analogue scale between the patients with ID and normal iron status. However, patients with ID type II reported the lowest appetite compared to patients with ID type I and Normal Iron (mean 5.4 ± 2.4 vs. 6.8 ± 2.1 vs. 6.7 ± 2.1, *p* = 0.05, respectively, Table 1).

#### 3.1.4. Regression Analyses of Handgrip Strength at Baseline

Mean HGS in female patients was 21 ± 9 kg and in male patients was 36 ± 11 kg. Notably, those patients with ID and TSAT < 20% showed more often a lower HGS in comparison to patients with ID and TSAT ≥ 20% (64% vs. 36%, *p* = 0.04).

In univariable logistic regression, low HGS was associated with the presence of ID, ID type II, TSAT < 20%, and age in the whole patients’ cohort (Table 2). Low HGS remained associated with ID type II after adjustment for BMI (Model 1). When analyzing low iron status only by marker TSAT < 20% low HGS was independently associated with low iron status after multivariable adjustment for BMI, age, and inflammation (Model 2).

### 3.2. Follow Up

One year after stroke (mean 382 ± 27 days), only 64 patients (46%) participated in the FU examination (FU cohort). Thus, 16 patients (11%) had problems traveling to the study center, 14 patients (10%) declined to continue the study, 6 patients (4%) had died, and 40 patients (29%) were lost to FU.

Comparing the entire study cohort with the FU cohort showed no difference in demographic, clinical, and biochemical parameters; stroke type and severity; comorbidities; and medication (Appendix A Appendix A).

#### 3.2.1. Iron Status

At baseline, ID was found in 31 patients (52%) of the FU cohort (Figure 1B). This proportion increased up to 77% at FU examination (*p* = 0.001, Figure 1B). According to the ID subtypes, the proportion of patients with ID type I in this cohort increased from 36% to 49% and patients with ID type II from 13% to 28% (*p* < 0.05) from baseline to FU, respectively. All of the female patients that participated in the FU examination (38%) were found to have ID. 

#### 3.2.2. Muscle Strength

Patients with ID remained with lower HGS at FU (mean 28.8 ± 12.0 vs. 44.5 ± 8.4 kg, *p* < 0.001, Figure 3A) and in a sensitivity analysis performed in male patients only (mean 37.1 ± 9.9 vs. 44.5 ± 8.4 kg, *p* < 0.05).

While patients with normal iron status showed an improvement of HGS at FU compared to baseline (mean 40.6 ± 8.3 vs. 44.0 ± 8.6 kg, *p* < 0.05), no improvement was observed in patients with ID (mean 29.3 ± 10.2 vs. 28.8 ± 12.0 kg, n.s., Figure 3B). The improvement of HGS was associated with normal iron status at FU (OR 3.7 [95% CI 1.01–13.5], *p* < 0.05).

#### 3.2.3. Nutritional Status

Patients who developed ID at FU were found to have reduced appetite at baseline (mean 6.4 ± 2.1 vs. 7.6 ± 1.6, *p* = 0.05). At FU, patients with ID had more frequently low nutritional status compared to patients with normal iron status (49% vs. 20%, *p* < 0.05).

#### 3.2.4. Logistic Regression Analyses of Handgrip Strength at Follow Up

In this restricted number of patients in the FU cohort, we found a trend in association between low HGS and reduced iron status. In invariable logistic regression analyses, ID, ID II, and TSAT < 20% all showed trends towards an association with low HGS (Table 3).

## 4. Discussion

Up to date, ID in patients with stroke has not been recognized as relevant clinical complication that might affect the clinical outcome. This study shows a high prevalence of ID, and an association with low muscle strength in patients with acute ischemic stroke. The main findings of this study are as follows: (1) the prevalence of ID was high in patients with acute ischemic stroke and even increased within one year after the acute event; (2) patients with ID had lower muscle strength after stroke; (3) patients with ID remained with lower muscle strength one year after stroke; and (4) while patients with normal iron status showed an improvement in muscle strength at FU, such an improvement of muscle strength was not observed in patients with ID.

The present study investigated the longitudinal changes in ID in patients after stroke. We observed no spontaneous improvement of iron status within one year after stroke but rather an increase in ID prevalence. To date, we lack longitudinal studies on development of ID in patients after stroke. The majority of clinical studies investigating ID in patients with acute and chronic diseases report ID prevalence obtained as a cross-sectional value only. Previously, a longitudinal multicenter clinical trial investigating an association between ID and unspecific inflammation in otherwise healthy adults showed a 12% increase to 39% in prevalence of ID within three years in community-dwelling older individuals [24]. Previous observations in patients with acute conditions, including acute exacerbation of chronic obstructive pulmonary disease, acute coronary syndrome, or acute HF, reported a prevalence of ID ranging between 20% and 80% [25,26,27]. In patients with chronic conditions, such as chronic heart failure, pulmonary arterial hypertension, or cancer, the prevalence of ID ranged between 30% and 50% [13,28,29,30]. Therefore, our results fit into the range of ID prevalence described for other chronic conditions. However, further studies are warranted to validate our observations.

### 4.1. ID Categories in Stroke

We investigated the distribution of ID categories in patients with acute and chronic stroke since a difference in the prevalence of both ID subtypes has been found in cardio-oncology patients and patients with acute and chronic heart failure [21,27,29,31]. In the present study, ID type I, defined by ferritin levels < 100 suggesting depleted iron stores, was most prevalent. The proportion of patients with this type of ID increased by 12% within one year of stroke. The main causes of iron store depletion are excessive blood loss and inadequate dietary iron uptake or absorption [11,12]. Acute blood loss in the context of acute ischemic stroke is not considered relevant for exploration of the ID in these patients. Since ID might develop as a consequence of malnutrition [31], we investigated appetite and nutritional status of patients in acute and chronic stroke. Dysphagia and eating-related difficulties due to disability are frequently observed after stroke [32,33]. None of the patients included in our study had eating difficulties secondary to stroke event. However, patients who developed ID during FU were found to have the lowest appetite in acute stroke and showed the lowest nutritional status upon FU examination. In addition, prescribed medications, including aspirin and proton pump inhibitors, might contribute to the depletion of iron stores [12]. Thus, in renal transplant recipients it was observed that the use of proton pump inhibitors was associated with ID and low ferritin levels independently of potential confounders [34]. In our study, almost 90% of patients in the FU cohort received aspirin for the secondary stroke prevention, and more than 20% of patients received proton pump inhibitors, which may explain the high prevalence of ID type I in the present study.

About one third of the patients with ID were found with ID type II, which is characterized by impaired iron metabolism triggered by systemic inflammation [35]. Indeed, we observed in these patients elevated C-reactive protein serum levels and white blood cell counts, indicating inflammation. We also found these patients to have reduced appetite and lower nutritional status one year after stroke.

In the present study, ID type I was found as a predominant mechanism of ID in patients with stroke. This is in line with previous reports, which observed this type of ID as a more frequent mechanism of ID in patients with cardio-oncologic diseases and patients with heart failure [27,28,36,37]. A recent prospective multicenter trial investigating about 700 patients with acute HF reported a predominance of ID type I and its persistence after treatment of acute HF [37]. The causes leading to the onset of ID type I in chronic stroke might include an inadequate iron uptake due to dietary habits and low appetite, as well as intake of medications, e.g., aspirin and proton pump inhibitors. ID type II is commonly observed in inflammatory condition in both clinical and experimental settings [11,12]. This type of ID is also expected in patients with underlying chronic comorbidities [12]. Our study is in line with these reports, as patients with inflammatory levels more often had ID type II, whereas patients without inflammation more often had ID type I. The therapeutic consequences of ID treatment depend on the underlying mechanism. While for the treatment and prevention of iron deficiency with low tissue iron stores (type I), parenteral iron supplementation, improved nutrition, and treatment of malabsorption may be considered, patients with ID type II should be treated for inflammation first [12]. Further investigations exploring the mechanisms of ID and the effects of iron supplementation on muscle recovery and outcome in acute and chronic stroke should be carried out.

### 4.2. Physical Performance and Muscle Strength

Lower physical performance has been reported in patients with ID and chronic diseases [6,7,8]. Patients with ID in the present study had a lower muscle strength assessed by handgrip and quadriceps muscle strength tests. Importantly, stroke severity according to the NIHSS and mRS was similar in patients with ID and without ID. ID patients with TSAT < 20% more often had low HGS. Notably, this cutoff has been shown to reflect ID in the bone marrow as assessed by bone marrow biopsies in patients with chronic HF [38]. TSAT < 20% is considered as a pathophysiological marker of reduced peripheral iron availability in all organs, including cardiac and skeletal muscle [39]. Our findings suggest that this could contribute to lower HGS in the present cohort.

We also found lower HGS in patients with ID at FU. Notably, we observed that patients without ID improved in HGS one year after stroke, whereas patients with ID did not show significant improvement.

The present results are consistent with the previous clinical trial demonstrating low physical capacity and worse clinical outcomes in the presence of ID during post-stroke rehabilitation [40].

### 4.3. Study Limitations

The present study has several limitations. This is a small-size prospective observational study including only patients with ischemic stroke with mild to moderate stroke severity. However, the study allowed the analysis of clinical and functional parameters in relation to ID, its prevalence, and different mechanisms of ID after stroke.

## 5. Conclusions

The present study showed that a significant proportion of patients with acute ischemic stroke present with iron deficiency. This ID is associated with lower muscle strength in acute and chronic stroke. ID might be underdiagnosed in patients with stroke. Assessment of iron status should be regularly performed in patients with stroke in order to diagnose and treat ID.

## Figures and Tables

**Figure 1 jcm-11-00595-f001:**
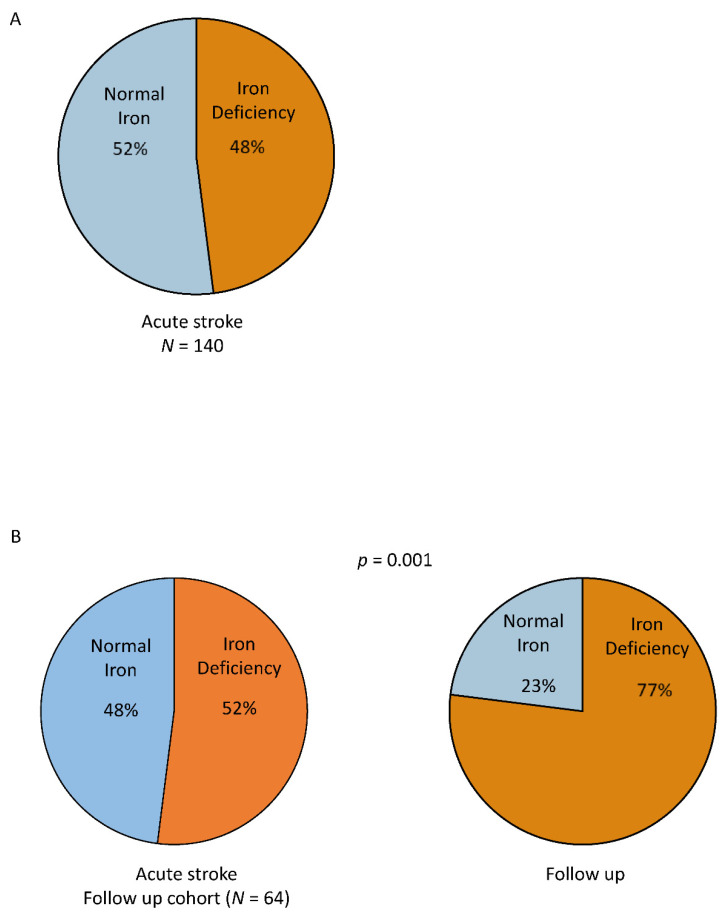
The prevalence of iron deficiency in acute ischemic stroke assessed at baseline (**A**) and in the follow up cohort at baseline and at one-year follow up (**B**).

**Figure 2 jcm-11-00595-f002:**
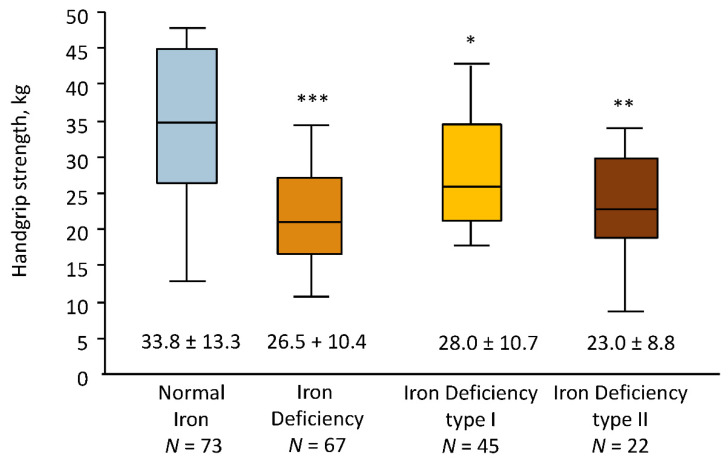
Handgrip strength at baseline in the study cohort divided into groups according to the presence of iron deficiency: normal iron status (normal iron), iron deficiency (ID), and categories of iron deficiency (ID type I and ID type II). * *p* < 0.05 vs. Normal Iron; ** *p* < 0.01 vs. Normal Iron; *** *p* < 0.001 vs. Normal Iron.

**Figure 3 jcm-11-00595-f003:**
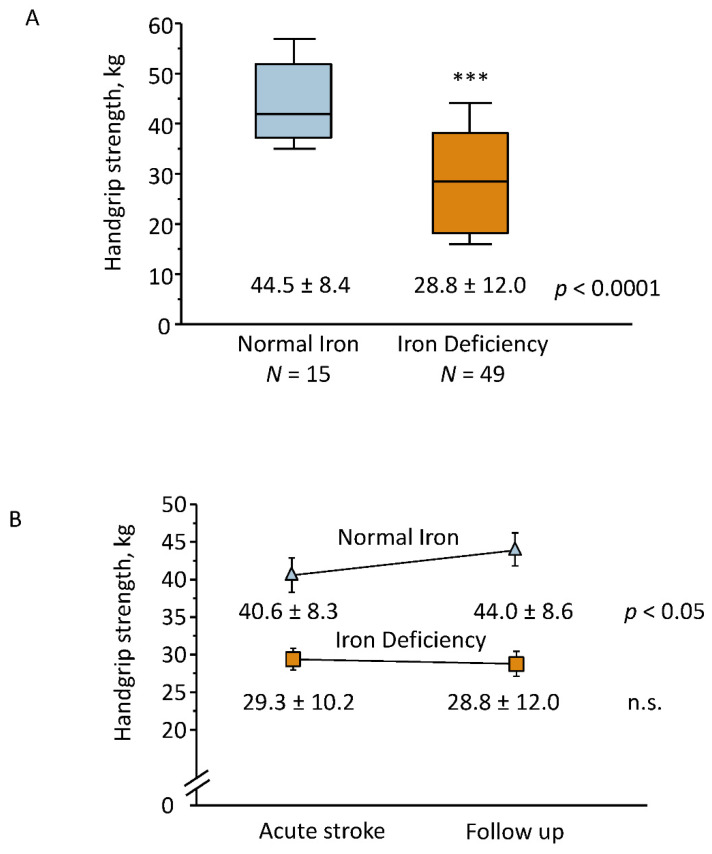
Handgrip strength at one-year follow up in patients with normal iron status (Normal Iron) and with iron deficiency (**A**). Changes in handgrip strength in patients with normal iron status (Normal Iron) and with iron deficiency (ID) within one year of follow up (**B**). n.s., non-significant. *** *p* < 0.001 vs. Normal Iron.

**Table 1 jcm-11-00595-t001:** Baseline characteristics of study population.

Clinical Parameters	All Patients *N* = 140	Normal Iron *N* = 73	ID *N* = 67	ID Type I *N* = 45	ID Type II *N* = 22	*p* Value ID vs. Normal Iron	*p* Value ID I vs. ID II vs. Normal Iron
Age, y, mean ± SD	69 ± 13	67 ± 12	70 ± 14	66 ± 14	77 ± 12 *	n.s.	<0.01
Body mass index, kg/m², mean ± SD	27.7 ± 4.6	28.3 ± 4.8	27.1 ± 4.2	26.0 ± 3.8	27.5 ± 4.3	n.s.	n.s.
Systolic RR, mmHg, mean ± SD	136 ± 28	137 ± 33	135 ± 21	146 ± 18	136 ± 22	n.s.	n.s.
Diastolic RR, mmHg, mean ± SD	79 ± 14	81 ± 13	77 ± 14	84 ± 10	75 ± 16	n.s.	n.s.
Female sex *N*, %	55 (39)	17 (23)	38 (57)	26 (58)	12 (55)	<0.001	<0.001
Self-reported appetite	6.5 ± 2.2	6.7 ± 2.1	6.3 ±2.3	6.8 ± 2.1	5.4 ± 2.4 *	n.s.	0.05
Stroke severity National Institute of Health Stroke Scale (NIHSS)							
Mean score ± SD	4.7 ± 3.4	4.8 ± 3.6	4.6 ± 3.1	4.1 ± 2.7	5.6 ± 3.6	n.s	n.s.
0–4, *N* (%)	83 (59)	42 (57)	41 (62)	32 (71)	9 (41)	n.s.	n.s.
Trial of ORG 10172 in Acute Stroke Treatment							
Cardioembolic, *N* (%)	44 (31)	20 (27)	24 (36)	12 (27)	12 (55) *	n.s.	<0.05
Large-artery atherosclerosis, *N* (%)	49 (35)	26 (36)	23 (35)	16 (36)	7 (32)	n.s.	n.s.
Small-vessel occlusion, *N* (%)	25 (18)	14 (19)	11 (16)	9 (20)	2 (9)	n.s.	n.s.
Stroke of undetermined etiology	22 (16)	13 (18)	9 (13)	8 (17)	1 (4)	n.s.	n.s.
Physical status							
Modified Rankin Scale (mRS)							
Mean score ± SD	2.4 ± 1.5	2.4 ± 1.6	2.4 ± 1.6	2.1 ± 1.3	2.9 ± 1.6	n.s	n.s.
0–1, *N* (%)	58 (41)	32 (44)	26 (39)	19 (42)	7 (32)	n.s.	n.s.
Low Handgrip strength, *N* (%)	61 (44)	28 (38)	33 (49)	19 (42)	14 (64)	n.s.	n.s.
Comorbidities							
Diabetes mellitus, *N* (%)	40 (29)	19 (26)	21 (38)	13 (36)	7 (33)	n.s.	n.s.
Arterial hypertension, *N* (%)	96 (69)	49 (67)	47 (70)	31 (69)	16 (76)	n.s.	n.s.
Dyslipidemia, *N* (%)	45 (32)	22 (30)	23 (34)	17 (38)	6 (27)	n.s.	n.s.
Anemia, *N* (%)	25 (19)	7 (10)	18 (27)	9 (20)	9 (41)	<0.01	<0.01
Cardiovascular disease, *N* (%)	56 (40)	29 (40)	29 (43)	15 (33)	5 (23)	n.s.	n.s.
Biochemistry							
Hemoglobin, mg/dL, mean ± SD	14.0 ± 1.9	14.6 ± 1.7	13.3 ± 1.8	13.4 ± 1.8	13.1 ± 1.9	n.s.	0.0001
White blood cells count	8.3 ± 2.5	8.1 ± 1.9	8.7 ± 3.1	8.1 ± 2.3	10.1 ± 4.2 **	n.s.	<0.01
Creatinine, mg/dL, mean ± SD	1.0 ± 0.4	1.0 ± 0.4	1.0 ± 0.4	1.0 ± 0.4	0.9 ± 1.9	n.s.	n.s.
Cholesterol, mg/dL, mean ± SD	186 ± 43	189 ± 37	182 ± 49	190 ± 50	166 ± 43 *	n.s.	n.s.
High density lipoprotein, mg/dL, mean ± SD	49 ± 15	46 ± 12	51 ± 16	54 ± 17 *	46 ± 12	<0.05	<0.05
Low density lipoprotein, mg/dL, mean ± SD	110 ± 38	115 ± 34	104 ± 41	109 ± 39	93 ± 45	n.s.	n.s.
Hemoglobin A1c, %, median [IQR]	5.9 [5.4–6.5]	5.8 [5.4–6.5]	5.9 [5.5–6.5]	5.8 [5.5–6.7]	5.6 [5.6–6.6]	n.s	n.s.
C-reactive protein, mg/L, median [IQR]	4.8 [1.7–11.8]	4.1 [1.7–12]	6.6 [1.7–10.35]	4.1 [1.7–7]	16.7 [7.2–26.2] *	n.s	<0.01
Systemic inflammation, *N* (%)	63 (45)	30 (41)	33 (49)	16 (35)	17 (77) *	n.s.	n.s.
Medication							
Antiplatelet drugs, *N* (%)	120 (86)	61 (84)	59 (88)	41 (91)	18 (82)	n.s.	n.s.
Anticoagulants, *N* (%)	29 (21)	16 (22)	13 (19)	6 (13)	7 (32)	n.s.	n.s.
Proton pump inhibitors, *N* (%)	34 (24)	16 (22)	18 (27)	13 (29)	5 (23)	n.s.	n.s.
β-blocker, *N* (%)	57 (41)	28 (38)	29 (43)	17 (38)	12 (55)	n.s.	n.s.
ACE-inhibitors, *N* (%)	65 (46)	39 (53)	26 (39)	15 (45)	11 (50)	n.s.	n.s.
Ca^2+^-channel antagonists, *N* (%)	14 (10)	7 (10)	7 (19)	5 (11)	2 (9)	n.s.	n.s.
Angiotensin II receptor blockers, *N* (%)	4 (3)	2 (3)	2 (3)	1 (2)	1 (5)	n.s.	n.s.
Diuretics, *N* (%)	29 (21)	12 (16)	17 (25)	10 (22)	7 (32)	n.s.	n.s.
Statins, *N* (%)	100 (71)	53 (73)	47 (70)	37 (82)	10 (45)	n.s.	n.s.

ACE, angiotensin converting enzyme; IQR, interquartile range; LDL, high-density lipoprotein; SD, standard deviation. * *p* < 0.05 vs. Normal Iron; ** *p* < 0.01 vs. Normal Iron. n.s., non-significant.

**Table 2 jcm-11-00595-t002:** Logistic regression analyses applying presence of handgrip strength below mean as a dependent variable at baseline.

Parameter	OR	95% CI	*p*	OR	95% CI	*p*	OR	95% CI	*p*
	Univariate			Model 1			Model 2		
Transferrin saturation < 20%	3.81	1.74–8.33	<0.001				3.0	1.24–7.18	<0.05
Presence of ID	2.04	1.00–4.15	<0.05						
Presence of ID I	0.96	0.44–2.08	0.9						
Presence of ID II	4.42	1.25–15.65	0.02	4.35	1.23–15.45	0.03			
BMI (per kg/m^2^ increase)	1.07	0.98–1.16	0.1	0.96	0.88–1.04	0.3	0.97	0.88–1.06	0.5
Age (per year increase)	1.07	1.04–1.11	<0.001				1.06	1.03–1.10	<0.001
NIHSS (per point increase)	1.06	0.95–1.19	0.3						
Hemoglobin, per mg/dL	0.88	0.72–1.07	0.2						
Presence of Inflammation	1.89	0.92–3.86	0.08				1.16	0.51–2.64	0.7

BMI, body mass index; ID, iron deficiency; NIHSS, National Institute of Health Stroke Scale. CI, confidence interval; OR, odds ratio.

**Table 3 jcm-11-00595-t003:** Univariable logistic regression analyses applying low handgrip strength as a dependent variable at FU.

Parameter	OR	95% CI	*p*
	Univariate		
Transferrin saturation < 20%	2.86	0.97–8.42	0.06
Presence of ID	3.0	0.91–9.91	0.07
Presence of ID I	2.59	0.73–9.25	0.1
Presence of ID II	3.9	0.91–16.8	0.07
BMI (per kg/m^2^ increase)	0.92	0.82–1.04	0.2
Age (per year increase)	1.08	1.02–1.14	<0.001
Hemoglobin, per mg/dL	0.98	0.75–1.29	0.9
Presence of Inflammation	4.69	0.94–23.3	0.06

BMI, body mass index; ID, iron deficiency.

## Data Availability

The data presented in this study are available on request from the corresponding author. The data are not publicity available due to privacy.

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
