# Peer review of "Iron Deficiency and Reduced Muscle Strength in Patients with Acute and Chronic Ischemic Stroke"

_jcm, 2022, doi:10.3390/jcm11030595_

Round 1
Reviewer 1 Report
The aim of the study was to «evaluate a prevalence of Iron Deficiency (ID) and to determinate its impact on long-term functional outcome in patients with ischemic stroke». This problem is actual for clinical medicine.
It is an important and high quality manuscript. But at the same time there are few nonessential critical comments and constructive remarks to authors.
- Baseline study examinations were performed in 140 patients with ischemic stroke, but after one year of follow-up – only in 64 (45.7%) of them. There is no information about the number of patients who died in the hospital or during the post-hospital follow-up period. So, it is not possible to evaluate the lost to follow-up proportion for survivals. It is important to add this information in the text of this manuscript. If this information is not available it must be described as the limitation of the study.
- Two groups were compared (140 and 64 patients). In this case it is not absolutely correct to write the next: «One year after stroke, prevalence of ID even increased up to 77%». It will be more correct to compare this parameter for the same 64 patients at baseline and after one year of follow-up.
- There are some studies that proved the «considerable impact of ID on prognosis». In this study the prevalence of ID was compared in 140 patients initially and only in 64 of them after one year of follow-up. The impact of ID on functional outcome was not estimated by special statistical analysis (for instance, with estimation of Odds Ratio and 95% CI). So, it will be better to modify the aim of study and exclude the words «to determinate impact».
So, the manuscript must be minimally revised without additional review.
Author Response
Reviewer 1
We are thankful to the reviewer for her/his appreciated and constructive comments.
- Baseline study examinations were performed in 140 patients with ischemic stroke, but after one year of follow-up – only in 64 (45.7%) of them. There is no information about the number of patients who died in the hospital or during the post-hospital follow-up period. So, it is not possible to evaluate the lost to follow-up proportion for survivals. It is important to add this information in the text of this manuscript. If this information is not available it must be described as the limitation of the study.
Replay: We thank the reviewer for this important comment. We provide now information regarding the lost to follow up.
One year after stroke, only 64 patients (46%) participated at the 1y FU examination (382 ± 27 days). Thus, 16 patients had problems to travel to the study centre, 14 patients declined to continue the study, 6 patients died, and 40 patients have not been attained.
This information has been added to the Results section of the manuscript.
- Two groups were compared (140 and 64 patients). In this case it is not absolutely correct to write the next: «One year after stroke, prevalence of ID even increased up to 77%». It will be more correct to compare this parameter for the same 64 patients at baseline and after one year of follow-up.
Replay: We thank the reviewer for this important comment. We clarified this by expressing that the increase reported relates only to those patients with a successful follow up 1 year later.
At baseline, ID was found in 31 patients (52%) of the later FU cohort (Figure 1B). This proportion increased up to 77% at 1yFU examination (p=0.001, Figure 1B). According to the ID subtypes, the proportion of patients with ID type I in this cohort increased from 36% up to 49% and patients with ID type II from 13% up to 28% (p<0.05) from baseline to 1 year FU, respectively.
Figure 1. (B) The prevalence of iron deficiency in the follow-up cohort at baseline and at 1 year follow-up.
- There are some studies that proved the «considerable impact of ID on prognosis». In this study the prevalence of ID was compared in 140 patients initially and only in 64 of them after one year of follow-up. The impact of ID on functional outcome was not estimated by special statistical analysis (for instance, with estimation of Odds Ratio and 95% CI). So, it will be better to modify the aim of study and exclude the words «to determinate impact».
Replay: We thank the reviewer for this remark. We modified now title, abstract and conclusion
Iron Deficiency and reduced Muscle Strength Recovery in Patients with Acute and Chronic Ischemic Stroke
Abstract: The aim of this prospective study was to evaluate a prevalence of ID and to evaluate its association with long-term functional
Conclusions: The present study shows that a significant proportion of patients with acute ischaemic stroke present with iron deficiency. This ID is associated with reduced muscle strength in acute and chronic stroke. ID might be underdiagnosed in patients with stroke. Assessment of iron status should be regularly performed in patients with stroke in order to prevent and treat ID.
stroke are should be carried out.

Reviewer 2 Report
Major comments
I commend the authors on the completion of the study. While the premise of the study is an important one in the management of stroke patients, I do have reservations about the validity of the findings due to potential confounders. Studies have shown that patients’ age, history of diabetes, Afib and other comorbidities, functional status prior to stroke, and motivation are predictive factors of functional progress assessed by muscle strength post-stroke (PMID: 25074799; PMID: 31610763).
Though the differences in the prevalence of co-morbidities among the categories of patients were not significantly different, the authors could not adjust for other important predictors such as motivation and functional status prior to stroke.
Another major limitation is the small sample size of this study. The authors also mentioned that the ID prevalence of 77% seen in this study was higher compared to other chronic conditions such as chronic heart failure, pulmonary arterial hypertension, and cancer. However, I would interpret this finding with caution due to the relatively smaller sample size of the study compared to other comparable chronic conditions.
Minor comments
The authors should consider making the title more specific. Consider changing it to ‘The effect of IDA on Muscle Strength Recovery in Patients with acute and chronic ischemic stroke’.
The authors mentioned that 64 patients were available for the 1y FU examination. It will be beneficial to know what happened to the other 76 patients. Were they lost to follow up…., did they die…...?
It is interesting that the authors found that the majority of patients had ID type I, characterized by depleted iron stores. Expectedly, Type II ID should have been more prevalent because most patients with ischemic stroke have chronic comorbidities. It will be helpful if the authors can discuss more in-depth their hypothesis/theory for these unusual differences. Are there other studies in the literature with similar or contrasting findings specifically on stroke in the literature? Such granular details in the discussion will make the manuscript more robust.
Author Response
Reviewer 2
We are thankful to the reviewer for her/his appreciated and constructive comments.
Major comments
I commend the authors on the completion of the study. While the premise of the study is an important one in the management of stroke patients, I do have reservations about the validity of the findings due to potential confounders. Studies have shown that patients’ age, history of diabetes, Afib and other comorbidities, functional status prior to stroke, and motivation are predictive factors of functional progress assessed by muscle strength post-stroke (PMID: 25074799; PMID: 31610763). Though the differences in the prevalence of co-morbidities among the categories of patients were not significantly different, the authors could not adjust for other important predictors such as motivation and functional status prior to stroke.
Replay: We thank the reviewer for this comment. Unfortunately, we were not able to collect the data regarding the motivation and functional status prior to the stroke. Therefore, and due to the small sample size of this pilot study, we were not able to adjust for these confounders. We will consider these analyses in our next clinical trials. However, we have measured iron deficiency at baseline in this stroke cohort. ID itself is known to cause fatigue, depression and lack of drive. This underlines the high need to timely diagnose and treat iron deficiency in order to improve key success factor such as motivation in stroke patients.
Another major limitation is the small sample size of this study. The authors also mentioned that the ID prevalence of 77% seen in this study was higher compared to other chronic conditions such as chronic heart failure, pulmonary arterial hypertension, and cancer. However, I would interpret this finding with caution due to the relatively smaller sample size of the study compared to other comparable chronic conditions.
Replay: We thank the reviewer for this comment. We agree with the concern of the small sample size, this has been addressed in the limitation section of the study. Indeed, high prevalence of ID in the present study compared to other chronic diseases might be due to the small sample. However, our study provides novel and clinically relevant information on a hitherto unaddressed topic. We therefore believe that the results are helpful to advance the clinical awareness for this problem. In accord we added to the discussion:
To best of our knowledge, the present study is the first to investigate the longitudinal changes of ID prevalence in patients with chronic stroke. In the present cohort, we observed 52% of patients with acute ischemic stroke suffering from ID, and the prevalence of ID increased by 25% within 1 year after acute event. To date, we lack longitudinal studies on development of ID over course of time in stroke patients. The majority of clinical studies investigating ID in patients with acute and chronic diseases report ID prevalence as a cross-sectional value only. Previously, a longitudinal multicenter clinical trial investigating an association between ID and unspecific inflammation in otherwise healthy adults showed a 12% increase of ID prevalence up to 39% within 3 years in community-dwelling older individuals [[1]]. Clinical studies investigating ID in patients with acute conditions, including acute exacerbation of COPD, acute coronary syndrome or acute heart failure, reported a prevalence of ID ranging between 20% and 80% [[2],[3],[4]]. In patients with chronic conditions such as chronic heart failure, pulmonary arterial hypertension or cancer, the prevalence of ID ranges between 30 and 50% [[5],[6],[7] [8]]. Further studies are warranted to validate our observations.
Minor comments
The authors should consider making the title more specific. Consider changing it to ‘The effect of IDA on Muscle Strength Recovery in Patients with acute and chronic ischemic stroke’.
Replay: We thank the reviewer for this comment. As the first reviewer suggested not to overinterpret our data as we did not adjust for factors like motivation of patients we had to change the title of the manuscript to:
“Iron Deficiency and reduced Muscle Strength Recovery in Patients with Acute and Chronic Ischemic Stroke”.
The authors mentioned that 64 patients were available for the 1y FU examination. It will be beneficial to know what happened to the other 76 patients. Were they lost to follow up…., did they die…...?
Replay: We thank the reviewer for this important comment. We provide now information regarding the lost to follow up.
One year after stroke, only 64 patients (46%) participated at the 1y FU examination (382 ± 27 days). Thus, 16 patients had problems to travel to the study centre, 14 patients declined to continue the study, 6 patients died, and 40 patients have not been attained.
This information has been added to the Results section of the manuscript.
It is interesting that the authors found that the majority of patients had ID type I, characterized by depleted iron stores. Expectedly, Type II ID should have been more prevalent because most patients with ischemic stroke have chronic comorbidities. It will be helpful if the authors can discuss more in-depth their hypothesis/theory for these unusual differences. Are there other studies in the literature with similar or contrasting findings specifically on stroke in the literature? Such granular details in the discussion will make the manuscript more robust.
Replay: We thank the reviewer for this very interesting comment. We add to the discussion:
In the present study, ID type I was found as a predominant mechanism of ID in patients with acute and chronic stroke. This is in line with previous reports which observed this type of ID as a more frequent mechanism of ID in patients with cardio-oncologic diseases and patients with heart failure [6,[9],[10],[11]]. A recent prospective multicentre trial investigating patients with acute HF reports a predominance of ID type I and its persistence after treatment of acute HF [11]. The causes leading to the onset of ID type I in chronic stroke might include an inadequate iron uptake due to dietary habits and low appetite as well as intake of medication e.g. aspirin and proton pump inhibitors. Type II ID is commonly observed in inflammatory condition in both, clinical and experimental settings [[12],[13]]. This type of ID is also expected in patients with underlying chronic comorbidities [13]. Our study is in line with these reports as patients with inflammatory levels had more often ID type II, whereas patients without inflammation had more preferably ID type I. However, we did not observe significant difference regarding the frequency of comorbidities except anaemia between patients with ID type I and type II. The therapeutic consequences of ID treatment depend on the underlying mechanism. While for the treatment and prevention of iron deficiency with low tissue iron stores (type I), an oral or parenteral iron supplementation, improved nutrition and treatment of malabsorption is suggested, in patients with ID type II underlying disease should be treated first [13]. Further investigations exploring the mechanisms of ID and the effects of iron supplementation on muscle recovery and outcome in acute and chronic stroke are should be carried out.
References
[1] Wieczorek M, Schwarz F, Sadlon A, Abderhalden LA, de Godoi Rezende Costa Molino C, Spahn DR, Schaer DJ, Orav EJ, Egli A, Bischoff-Ferrari HA; DO-HEALTH Research group. Iron deficiency and biomarkers of inflammation: a 3-year prospective analysis of the DO-HEALTH trial. Aging Clin Exp Res. 2021. doi: 10.1007/s40520-021-01955-3.
[2] Silverberg DS, Mor R, Weu MT, Schwartz D, Schwartz IF, Chernin G. Anemia and iron deficiency in COPD patients: prevalence and the effects of correction of the anemia with erythropoiesis stimulating agents and intravenous iron. BMC Pulm Med. 2014;14:24.
[3] Silva C, Martins J, Campos I, Arantes C, Braga CG, Salomé N, Gaspar A, Azevedo P, Álvares Pereira M, Marques J, Vieira C. Prognostic impact of iron deficiency in acute coronary syndromes. Rev Port Cardiol. 2021;40:525-536.
[4] Jacob J, Miró Ò, Ferre C, Borraz-Ordás C, Llopis-García G, Comabella R, Fernández-Cañadas JM, Mercado A, Roset A, Richard-Espiga F, Valero-Domènech A, Martínez-Gimeno JL, Martín-Sánchez FJ, Llorens P, Berrocal-Gil P, Pérez-Durá MJ, Álvarez-Pérez JM, López-Díez P, Herrero-Puente P, Comín-Colet J. ID and safety of ferric carboxymaltose in patients with acute heart failure. AHF-ID study. Int J Clin Pract. 2020;74:e13584.
[5] von Haehling S, Ebner N, Evertz R, Ponikowski P, Anker SD. ID in Heart Failure: An Overview. JACC Heart Fail. 2019;7:36-46.
[6] Jankowska EA, Rozentryt P, Witkowska A, Nowak J, Hartmann O, Ponikowska B, Borodulin-Nadzieja L, Banasiak W, Polonski L, Filippatos G, McMurray JJ, Anker SD, Ponikowski P. Iron deficiency: an ominous sign in patients with systolic chronic heart failure. Eur Heart J. 2010;31:1872-80
[7] Quatredeniers M, Mendes-Ferreira P, Santos-Ribeiro D, Nakhleh MK, Ghigna MR, Cohen-Kaminsky S, Perros F. ID in Pulmonary Arterial Hypertension: A Deep Dive into the Mechanisms. Cells. 2021;10:477
[8] Čiburienė E, Čelutkienė J, Aidietienė S, Ščerbickaitė G, Lyon AR. The prevalence of ID and anemia and their impact on survival in patients at a cardio-oncology clinic. Cardiooncology. 2020;6:29.
[9] Camaschella C. ID. Blood. 2019;133:30-39.
[10] Pozzo J, Fournier P, Delmas C, Vervueren PL, Roncalli J, Elbaz M, Galinier M, Lairez O. Absolute iron deficiency without anaemia in patients with chronic systolic heart failure is associated with poorer functional capacity. Arch Cardiovasc Dis. 2017;110:99-105.
[11] van Dalen DH, Kragten JA, Emans ME, van Ofwegen-Hanekamp CEE, Klaarwater CCR, Spanjers MHA, Hendrick R, van Deursen CTBM, Brunner-La Rocca HP. Acute heart failure and iron deficiency: a prospective, multicentre, observational study. ESC Heart Fail. 2021. doi: 10.1002/ehf2.13737.
[12] Guida C, Altamura S, Klein FA, Galy B, Boutros M, Ulmer AJ, Hentze MW, Muckenthaler MU. A novel inflammatory pathway mediating rapid hepcidin-independent hypoferremia. Blood. 2015;125:2265-75.
[13] Pasricha SR, Tye-Din J, Muckenthaler MU, Swinkels DW. Iron deficiency. Lancet. 2021;397:233-248.
